# Black Carbon and Its Effect on Carbon Sequestration in Soil

**Marek Kopecký** [1,*] , **Ladislav Kolář** [1] , **Radka Váchalová** [1] , **Petr Konvalina** [1] , **Jana Batt** [1] , **Petr Mráz** [1] , **Ladislav Menšík** [2] , **Trong Nghia Hoang** [1] and **Miroslav Dumbrovský** [3]

1. Faculty of Agriculture, University of South Bohemia in Ceske Budejovice, Studentska 1668, 37005 Ceske Budejovice, Czech Republic; kolar@zf.jcu.cz (L.K.); vachalr@zf.jcu.cz (R.V.); konvalina@zf.jcu.cz (P.K.); janabatt@volny.cz (J.B.); mrazpe01@zf.jcu.cz (P.M.); hoangn00@zf.jcu.cz (T.N.H.)
2. Crop Research Institute, Division of Crop Management Systems, Drnovska 507-73, 16106 Prague 6, Czech Republic; ladislav.mensik@vurv.cz
3. Faculty of Civil Engineering, Brno University of Technology, Zizkova 17, 60200 Brno, Czech Republic; dumbrovsky.m@vutbr.cz
* Correspondence: mkopecky@zf.jcu.cz

**Abstract:** The properties of black carbon (BC) are described very differently in the literature, even when determined by the same methodological procedure. To clarify this discrepancy, BC was investigated in the clay Cambisols of southern Bohemia, Czech Republic, in groups of soils with lower and higher deposition of its atmospheric fallout. The BC determination was performed according to a modified method of Kuhlbusch and Crutzen (1995). The amount of the free light fraction, the occluded light fraction of soil organic matter and its ratio, the amount of heavy soil fraction DF, and its soil organic matter DFOM were determined. Other soil characteristics were identified. It was found that there are two very different types of BC in soils. Historical BC from biomass fires, and new, anthropogenic, from the furnace and transport fumes. Historical BC has a significant effect on the organic matter of the heavy soil fraction, on the ratio of the free and occluded soil organic matter fraction, and the number of water-resistant soil aggregates. Anthropogenic BC does not have this effect. Because this form of BC is not significantly stabilized by the colloidal mineral fraction, it is necessary to take general data on BC's high stability and resistance to mineralization in the soil with circumspection.

**Keywords:** anthropogenic black carbon; density fractionation; historical black carbon; waterproof macro-aggregates

## 1. Introduction

Over the last 150 years, there has been a decrease in organic carbon ($C_{org}$) in the world's soils [1] and a consequent increase in atmospheric $CO_2$ [2]. Therefore, carbon sequestration in stable and resistant fractions of soil organic matter (SOM) in the soil is very current. The concept of carbon sequestration focuses on increasing primary production ($CO_2$ consumption) or reducing the rate of $CO_2$ production by SOM mineralization [3]. Long-term carbon sequestration in soil represents stable black carbon (BC) fractions [4]. Soil organic matter consists of functional pools that differ in their rate of decomposition. The labile part is the primary source of energy for soil microorganisms and contributes to the nutrient regime of soils [5]. This applies in particular to the water-soluble part of SOM. However, other pools are also important—semilabile, stable, and inert [6]. In the past, humic substances, humic acids, humins, and partially fulvic acids were considered stable forms of soil carbon. At present, this humification model is widely criticized, and the presumed stability of humic substances is questioned [7,8]. However, criticism of this conception has also emerged [9].

In any case, in the laboratory determination, the content of humic substances in the soil is "increased" during the determination by those black carbon fractions, which, similar to humic substances, can be extracted with an alkaline solution [10,11]. Therefore,

the humification model is replaced by the SOM fractionation, according to specific stabilization mechanisms. The protection of SOM against biodegradation is ensured by its spatial inaccessibility (in soil aggregates) and organic matter stabilization in the formed organomineral complexes [12–14]. Modern instrumental analysis methods (thermal analysis, nuclear magnetic resonance, and pyrolysis) have shown significant heterogeneity of humates in different soils and fractions of a single soil. Criticism of the humification model is therefore justified.

The structural features and chemical composition of highly aromatic soil humic acids (HA) suggest that these HA are derived from BC and not natural native plant materials. Humic acids from laboratory oxidized BC also show remarkable similarities in chemical compositions and spectroscopic data with highly aromatic soil humic acids. Therefore, BC is considered a possible source of the chemically most stable, aromatic soil carbon pool [15].

Black carbon is defined by Goldberg [16] as a mixed product that results from the incomplete combustion of fossil fuels, wood, and biomass, as well as from certain industrial processes, such as the production of carbon black for automobile tires and printing inks. He understood BC as a mixture of different charcoals, which he defines as a porous, solid product, containing 85–98% C, produced by carbonization of carbonaceous materials, at temperatures up to 600 °C in the absence of air. An overview of the chemical and physical properties of these substances is described by Mantell [17]. From the point of view of environmental relations, two characteristics of BC are important: high chemical stability at usual water, air, and soil temperatures, and high sorption activity, which depend on the original carbonaceous material, chemical characteristics of the environment, in which BC was formed and the reaction time and temperature during its arising [18]. Medalia and Rivin [19] distinguish four types of BC, according to particle size, their morphology, origin, and surface properties. Kuhlbusch and Crutzen [20] define BC as the fire produced carbon fraction with a molar H/C ratio of ≤0.2, which is resistant to heating to 340 °C in pure oxygen. The term BC includes several other terms: char, charcoal, soot, elemental carbon, pyrogenic carbon [21]. BC is considered an important sink in the global carbon pool [22]; it may represent an inert carbon pool used in SOM models [23].

It is estimated that the annual BC production from biomass combustion is 1 Tg [22], with total carbon content in the biosphere, atmosphere, and sea pools of $3 \times 10^6$ Tg. Thus, it is clear that BC destruction occurs, although it is generally considered extremely stable. Kuhlbusch et al. [24] give an estimate at 50–270 Tg per year, with more than 90% of BC coming from terrestrial ecosystems. There are a number of other estimates of BC production in the literature, but these are mostly geographically smaller units.

There are two destructive mechanisms of BC: photochemical and microbial decay. Human society releases other BC into the environment through the deliberate burning of forests and fossil fuels [4], the operation of engines vehicles, and the production of soot, graphite, and activated carbon. Recently, pyrolytic processes in the chemical industry have developed rapidly. Their solid waste, biochar, is another source of BC [25].

Most of this BC is stored in soils [26] and can form significant fractions of soil carbon [4,27]. The danger is the absorption of polycyclic aromatic hydrocarbons (PAHs) and other organic pollutants, sorb on BC [28]. Abiotic and microbial oxidation can form functional groups with a network of negative charges on the surface of BC particles [29]. A high concentration of COOH groups has been demonstrated after oxidative degradation of burned plants by dilute $HNO_3$ [30]. Influencing nutritional dynamics in soil has also been demonstrated [31,32]. The formation of carboxyl groups or other negatively charged groups in BC-added soils can be caused by two processes: (1) surface oxidation of BC particles themselves or (2) absorption of highly oxidized organic matter on the surface of BC particles [33]. The effect of BC oxidation on the increase in cation exchange capacity (CEC) has been demonstrated [34].

Analytical methods for the quantitative determination of BC in soils and other matrices are quite problematic due to the analyzed material's high heterogeneity and insolubility. An overview of basic methods is given by Goldberg [16]. Methods of spectroscopic, chemical

oxidation [35], oxidation to $CO_2$ after removal of other C-components [20], the combination of spectroscopy, and nuclear magnetic resonance [36] are used. It was found that in the oxidative degradation of coal, polycyclic and substituted aromatic centers are converted to benzenecarboxylic acids—BPCA [37]. This fact also applies to coal pyrolytic residues [38]. Schnitzer [39] found that BPCAs derived only from benzene rings unsubstituted by oxygen, but only C atoms. This is a feature typical of black carbon. That is why Glaser et al. [21] assumed that BPCA could be used as a specific measure of black carbon in soils.

Hedges et al. [40] published "The carbon combustion continuum of black carbon". It shows the carbon combustion products from large, reactive particles of weakly charred biomass through char, charcoal, soot, and graphite to submicroscopic black carbon particles, and outlines possible analytical determination methods. Chemical methods are only suitable for graphite, soot and charcoal; thermochemical methods only for graphite and soot [41]; visual methods only for weakly carbonized biomass [26]. The whole area of all charred biomass products can be captured only by CP/MAS 13C NMR. The most significant part of BC components (char, charcoal) can be captured by BPCA methods [21], molecular markers [42], and ultra-high mass spectrometry [26]. Combinations of these methods are also used [20].

Black carbon can affect the long-term storage of carbon in the soil in two ways—by its own stability and by its influence on two main mechanisms of physical stabilization of organic matter in the soil, which is physical protection in soil aggregates and stabilization by the formation of organomineral complexes [12]. Macro-aggregates (>250 μm) have a higher concentration of SOM than micro-aggregates [43] because macro-aggregates contain more binders [44]. Decomposed organic material is also a transitional sealant [45]. However, tillage management also has importance here [46].

Ultrasonic dispersion of stable aggregates and determination of density allows separation into three different fractions according to various physical protection mechanisms [47]. The free SOM fraction (FF) is isolated before ultrasonic dispersion, the occluded SOM fraction (OF) is after ultrasonic dispersion. The last fraction is organomineral, so-called "heavy" (DF). Its organic matter is called DFOM. The FF fraction is labile, the OF fraction more stable. The DF fraction is very stable, with a mineralization period of decades to centuries [12,48,49]. However, even the labile FF fraction can be further divided by density fractionation. It is divided into a free light fraction of organic matter (fLFOM), which is very labile, and a more stable occluded light fraction (oLFOM). The fLFOM fraction is on the surface of the aggregates; the oLFOM fraction is inside the aggregates [50]. For oLFOM to mineralize, it must first be released from the aggregates. Therefore, oLFOM is more stable the more considerable the number of water-resistant aggregates is in the soil [51].

However, BC particles are also part of the light organic matter fraction (LFOM). Therefore, the data on the ratio of fLFOM to oLFOM are essential in studying the stability of BC in soil [52].

The work aimed to determine the relationship between the amount and properties of black carbon in the clay Cambisols of South Bohemia, Czech Republic, to the amount of water-resistant macro-aggregates, the ratio of the free occluded light fraction of organic matter and the amount of heavy fraction of organic matter. According to the results, to contribute to the discussion of whether carbon sequestration in the form of BC is more or less significant in the given soil and climatic conditions at medium altitude and in a relatively clean landscape.

## 2. Materials and Methods

### 2.1. Soil Samples Processing

Cambisols are the predominant soil type in South Bohemia. The research focused on medium-heavy soils—loam soils. Ten localities were selected where the deposition of emissions from local heating plants and motor transport can be assumed. These localities were located east of relatively large human settlements; the exact places were selected based on maps of the prevailing wind direction. Namely: Tábor, Písek, Strakonice, Prachatice,

3 × České Budějovice, 2 × Český Krumlov, Jindřichův Hradec. Another ten samples were taken from localities relatively remote from sources of anthropogenic pollution; namely Křišťanov, Záhvozdí, Slavkov, Malšín, Jaroměř, Malonty, Pohorská Ves, Paseky, Hartunkov, Rychnov. The map of sampling localities is shown in Figure S1. Sampling was performed using a pedological sampling rod on arable land from a depth of 0–0.180 mm. Twenty samples were taken from each locality (from one particular land block) and composited. Soil samples were collected in May 2020. All analyses described below were performed six times.

The method described by Kuhlbusch and Crutzen [20] was used to determine BC. This method was originally intended to determine BC in the combustion residues of various vegetation types. The organic content is too low for soil analyses. This disadvantage was minimized by removing specifically heavier mineral particles using the pipetting method commonly used to determine soil granularity. Soil samples (sieved soil < 2000 μm) were first dispersed by boiling for 30 min with alkaline solution (dissolved 35.7 g of $(NaPO_3)_6$ and 7.94 g of $Na_2CO_3$ per liter of water). Particles larger than 250 μm were then removed through a sieve. The resulting suspension was diluted to a uniform volume (1000 mL) and, after mixing, transferred to a sedimentation device to determine soil granularity. The procedure was the same as used for determining soil granularity by the pipetting method. A fraction smaller than 50 μm was collected. Although this method of concentration increase of organic matter in the sample results in the loss of particulate organic matter of the soil, which is usually trapped on the sieve together with the sand, the transformed primary organic matter to which BC belongs remains in the most subtle soil fraction. A further increase in the concentration of the organic component was achieved by density fractionation with a NaI solution with a density of 1.6 g/cm$^3$. The isolated soil particles ≤50 μm in the centrifuge tube were shaken in an orbital shaker (170 rpm/min) with ten 5 mm diameter glass beads with 40 mL of 1.6 g/cm$^3$ sodium iodide solution for 18 h. After vacuum filtration of the supernatant, the free light fraction (fLF) and the occluded light fraction (oLF) and their soil organic matter fLFOM and oLFOM were obtained. These two fractions, according to Balesdent et al. [52], contain BC. The sediment in NaI solution is a heavy fraction (DF) with a density greater than 1.6 g/cm$^3$.

### 2.2. Black Carbon Content Determination

Elemental analyses (Vario EL CUBE) of dried and pulverized fLFOM and oLFOM samples made it possible to obtain data on the samples' total carbon (TC) and total hydrogen (TH). The aliquot proportion of the samples was sequentially extracted in a centrifuge tube with 1 M NaOH, 70% HNO$_3$, 1% HCl and twice with deionized water. After drying and weighing the residues, a second elemental analysis was performed to determine the total carbon (TC1) and hydrogen (TH1) content after extraction. It is then possible to calculate the removed carbon and hydrogen (inorganic carbon of carbonates IC, organic carbon released by solvent extraction OC1, and hydrogen removed by solvent extraction OH2).

A second purification step is needed to remove the residual organic carbon and hydrogen (OC2, OH2)—a thermal process. Parallel weighed samples of the pretreated material were exposed to a temperature of 340 °C for 2 h in a stream of oxygen (500 mL/min). The third elemental analysis will make it possible to determine black carbon and hydrogen bound to BC (BH) after deducting the volatilized carbon by the fire (VC). The molar H/C ratio (BH/BC) determined without correction for hydrogen, possibly bound to minerals (from OC2 and OH2), which was recommended by the cited authors [20].

### 2.3. Determination of the Structure of Soil

Determination of the structural condition of soil samples was performed by sieving and sedimentation [53]. Water-stable aggregates were fractionated into micro-aggregates (<250 μm) and macro-aggregates (>250 μm) according to John et al. [48]: 50 g of sieved soil (≤10 mm) was dried at 40 °C for 48 h and sieved in deionized water in a sieving machine for 10 min. After 50 vertical lifts of the sieve (38 mm), the water-stable aggregates

(>250 μm) were sprayed onto a vacuum filter, the water was aspirated and the aggregates dried at 40 °C. Particles that passed through the sieve (<250 μm) were isolated by the addition of 2.5 mL of 0.5 M AlCl$_3$ solution per 1000 mL of supernatant, after which they were decanted and dried for 48 h at 40 °C.

### 2.4. Determination of Free Light Fraction fLFOM, Occluded Light Fraction oLFOM, and Heavy Fraction DFOM of Soil Organic Matter

This determination was performed according to the method suggested by Balesdent et al. [54] and Golchin et al. [55]. The procedure for determining fLFOM was as follows: 10 g of sand-free soil samples (<2000 μm) was inserted into the centrifuge tube with 40 mL solution of sodium polytungstate at the density of 1.8 g/cm$^3$ (Sometu, Berlin, Germany). After shaking by hand, the suspension was allowed to stand for 30 min. Samples were then centrifuged (2000× $g$) for 30 min. The supernatant was vacuum filtered through a filter with pores <45 μm and then washed with 2000 mL of deionized water. After filtration, the amount of fLFOM was determined.

The remaining soil material in the centrifuge tube was then shaken on an orbital shaker (175 rpm/min) with ten 5 mm diameter glass beads with 40 mL of sodium poly-tungstate solution again at a density of 1.8 g/cm$^3$ for 18 h. After the first centrifugation and decantation, the soil particles were resuspended in 40 mL of polytungstate solution and centrifuged. This repeated procedure aims to completely separate the occluded light fraction oLF from the heavy fraction DF. The supernatants of both suspensions were combined and vacuum filtered. The remaining particles, corresponding to the heavy fraction DF with a density >1.8 g/cm$^3$, were washed with 1500 mL of deionized water to remove residual polytungstate. The suspension was then precipitated by adding 2.5 mL of 0.5M AlCl$_3$ per 1000 mL of supernatant and the supernatant was decanted with water. The heavy fraction DF was filtered off and washed with 500 mL of deionized water. This was followed by the carbon content determination in oLFOM and DFOM fractions.

### 2.5. Statistical Analysis

The data of Figures 1 and 2 were statistically evaluated by an analysis of variance (ANOVA) and the results were subsequently compared by a post-hoc Tukey HSD test. The method of principal component analysis (PCA) and factor analysis (FA) [56] was used for multivariate statistical analysis of measured data. Statistical analyses, including graphical outputs, were processed in STATISTICA (version 14, TIBCO Software, Inc., Palo Alto, CA, USA, 2021).

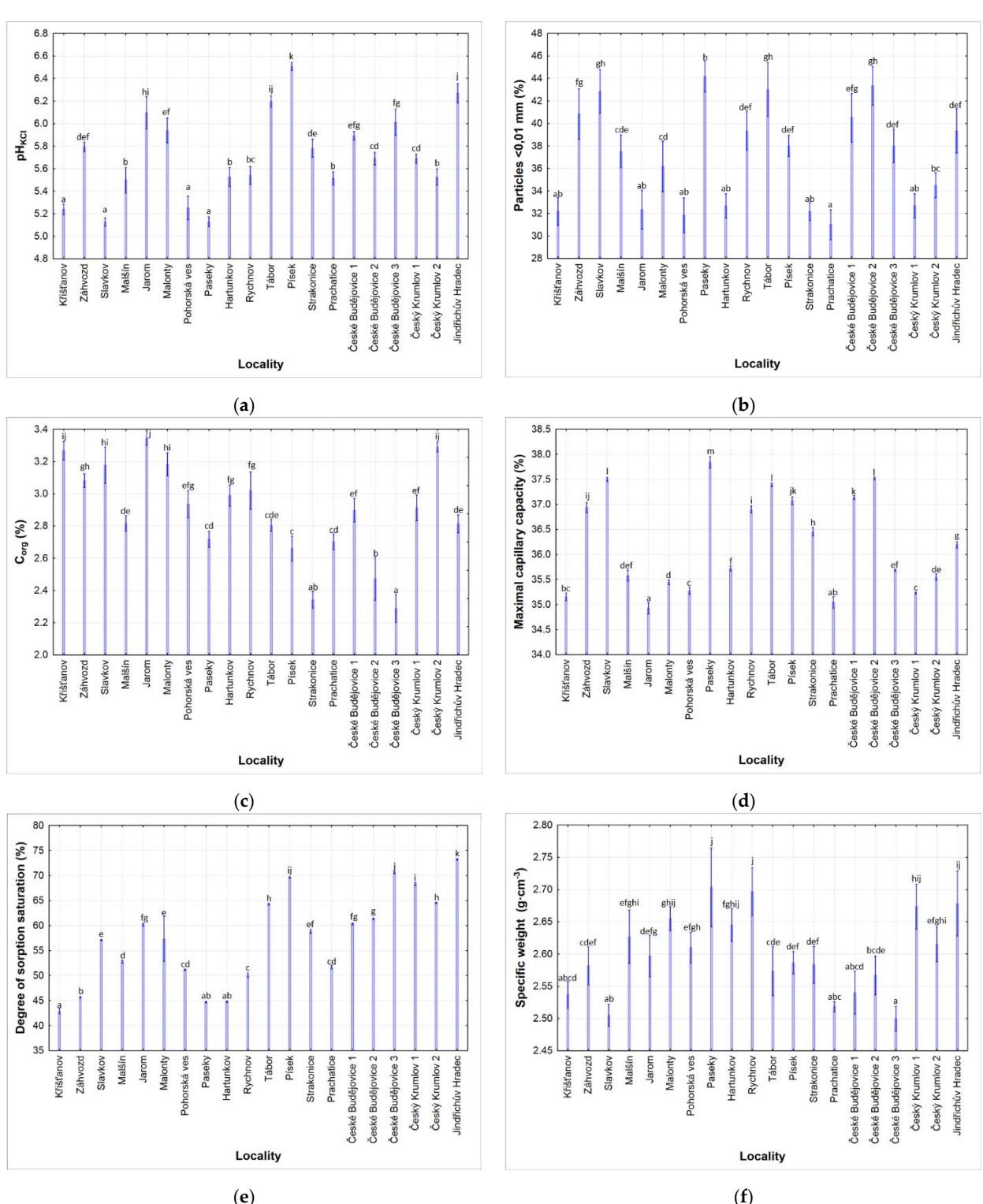

(**a**)

(**b**)

(**c**)

(**d**)

(**e**)

(**f**)

**Figure 1.** *Cont.*

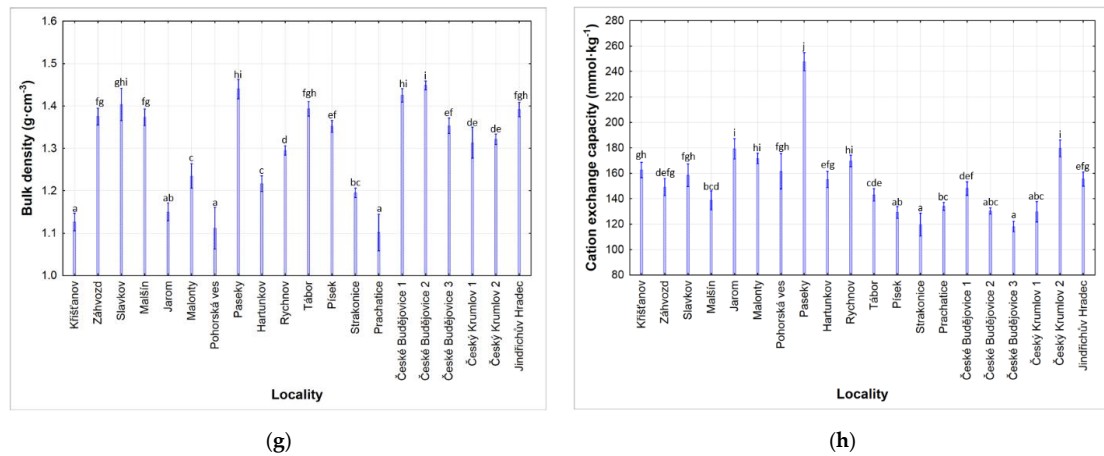

(**g**)            (**h**)

**Figure 1.** Analyses of samples of clay Cambisols (ANOVA, $p < 0.00001$): (**a**) $pH_{KCl}$ ($F_{(19, 100)} = 161.38$); (**b**) particles <0.01 mm ($F_{(19, 100)} = 48.726$); (**c**) $C_{org}$ ($F_{(19, 100)} = 111.06$); (**d**) maximal capillary capacity ($F_{(19, 100)} = 1136.8$); (**e**) degree of sorption saturation ($F_{(19, 100)} = 535.19$); (**f**) specific weight ($F_{(19, 100)} = 24.979$); (**g**) bulk density ($F_{(19, 100)} = 143.02$); (**h**) cation exchange capacity ($F_{(19, 100)} = 118.67$); significant differences between localities are shown in letters (Tukey's honest significance test; $p = 0.05$). Note: $C_{org}$—soil organic carbon (%).

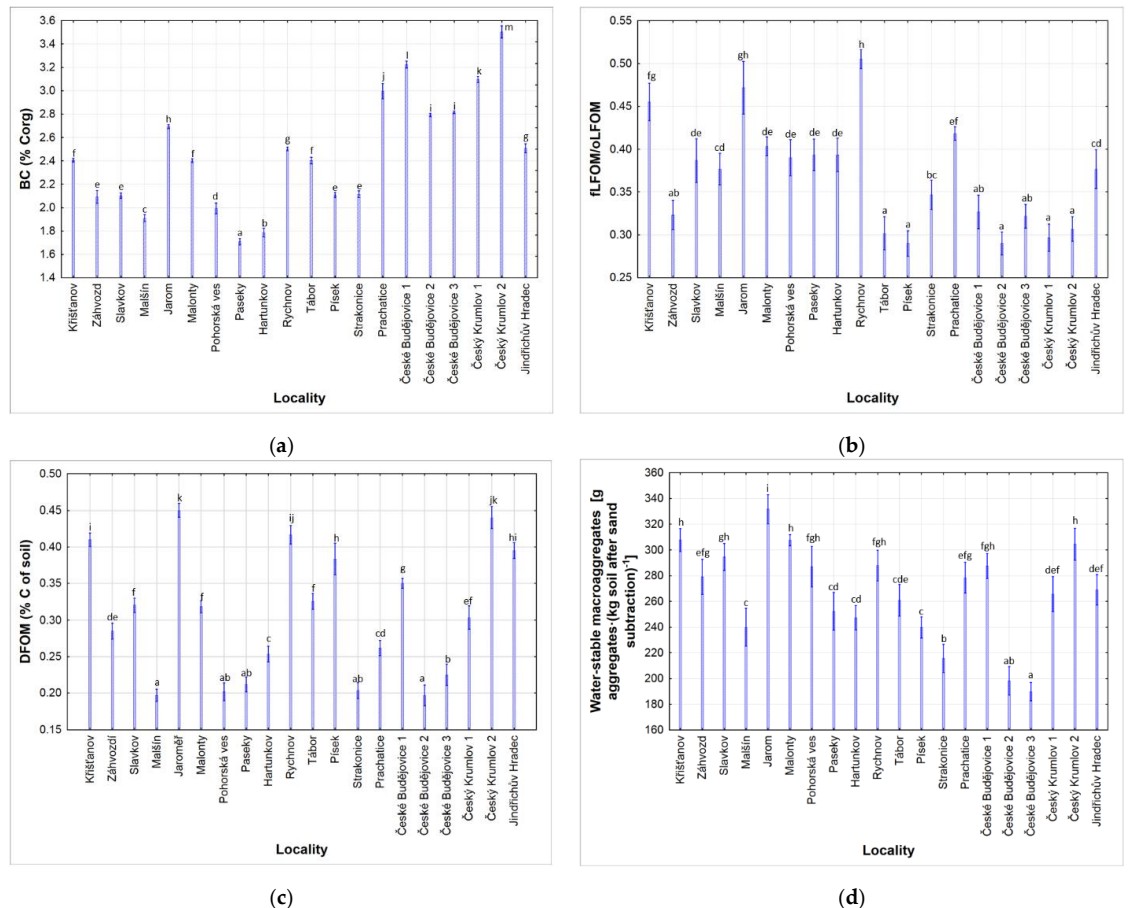

(**a**)            (**b**)

(**c**)            (**d**)

**Figure 2.** Amounts of soil organic matter fractions monitored in samples of clay Cambisols (ANOVA, $p < 0.00001$): (**a**) Black carbon; (**b**) the ratio of free light fraction and the occluded light fraction of soil organic matter; (**c**) the organic matter of the heavy soil fraction; (**d**) water-stable macroaggregates. Note: BC—black carbon ($F_{(19, 100)} = 1494.3$); fLFOM/oLFOM—the ratio of free light fraction and the occluded light fraction of soil organic matter ($F_{(19, 100)} = 76.991$); DFOM—the organic matter of the heavy soil fraction ($F_{(19, 100)} = 343.35$). Water-stable macroaggregates ($F_{(19, 100)} = 69.371$); significant differences between localities are shown in letters (Tukey's honest significance test; $p = 0.05$).

## 3. Results and Discussion

### 3.1. Soil Properties

The average results (including standard deviations) of all monitored soil characteristics are summarized in Table S1. Soil analysis proved that clay Cambisols from experimental localities corresponds to average values in the Czech Republic. The specific gravity of the examined soils is in the range of 2.48–2.76 g/cm$^3$, which corresponds to the surface humus horizons. The bulk density is slightly lower (1.05–1.47 g/cm$^3$) than would correspond to clay topsoil (1.34–1.40 g/cm$^3$). The content of clay particles <0.01 mm is in the range of 30–45% for all samples, which corresponds to clay soils. The maximum capillary capacity is slightly higher than that of topsoil of clay soils (35%), probably due to the relatively high content of $C_{org}$, which here reaches the category of "excellent" (2.9%). Cation exchange capacity should be high in these high carbon soils. However, for most samples, it is only medium (130–240 mmol/kg soil), probably due to the higher acidity of all samples, corresponding to the category of weakly acidic to acidic soils (5.1–6.5 $pH_{KCl}$). For the same reason, the degree of sorption saturation is only medium (50–75%) or even lower. The results of the analyses of soil samples are summarized in Figure 1.

### 3.2. Black Carbon Content

Important results concerning soil organic matter are summarized in Figure 2. Samples of soils from relatively cleaner localities in southern Bohemia showed on average lower BC content (2.16% $C_{org}$) than samples with higher deposition of atmospheric fallout (2.76% $C_{org}$). These samples from sites with higher atmospheric pollution probably have BC of anthropogenic origin (ABC), i.e., unburned residues of industrial and local furnaces and emissions from transport, especially soot from unadjusted diesel engines. The study of historical sources did not prove the existence of recorded large fires in any of the experimental sites. Thus, it is clear that the group of samples from "clean" sites contains ancient historical BC (HBC), which can be partially oxidized over a long period of storage in the soil. As a result, its properties, especially sorption and ion exchange, will differ from relatively young, anthropogenic BC soils from atmospherically contaminated areas. In general, however, it can be stated that the amount of BC found in the whole experiment is meagre.

The finding of low BC values could be caused either by the chosen analytical method or by the used purification operations. BC analysis is generally considered very problematic because the analyst does not have a chemical individual, but always a very varied mixture. Although there are modern methods for determining BC in soils [21,36], in addition to demanding instrumental techniques, it is always necessary to use indirect relationships to calculate BC content in soils. The method of Kuhlbusch and Crutzen [20] was used in this research because BC defines unambiguously. In our opinion, the cleaning operations used could affect the correctness of the results, but not the accuracy. As this is a comparative work, we believe that the final evaluation of the cleaning operation will not be affected.

The average amount of soil organic matter DFOM in the heavy fraction DF in the soils is a precondition for higher contamination by atmospheric fallout (0.3083%) almost identical to in soils less contaminated (0.3397%). However, its relation correlates with the BC content (r = 0.899268), and with the ratio of free light fraction and the occluded light fraction of soil organic matter (fLFOM/oLFOM) (r = 0.644752) in localities without deposition. In localities with deposition, the relationship to BC is not correlated. This is evidence that long-lived HBC is strongly bound to the mineral colloidal soil fraction and, of course, to the less stable LFOM. On the contrary, the "young" ABC has not yet stably bound to mineral colloids, either because ABC (from traffic, resulting from the combustion of fuels in furnaces), is less stable than HBC from biomass fires, or it has not yet oxidized in the soil [57]. Therefore, no negatively charged carboxyl sites and other functional groups have formed on its surface, and therefore this ABC has only a sorption ability, not an ion exchangeability. For the firm connection of the mineral colloidal clay fraction with the colloidal organic fraction, the ion-exchange capacity is more important than the sorption capacity. By analogy, soil

aggregates are much easier to form by combining mineral colloids with humic acid colloids than connecting them with soil non-humidified organic matter [13,14].

Anthropogenic BC, which does not affect DFOM in the heavy soil fraction of DF, forces the hypothesis that it will more easily undergo mineralization destruction in soil [22]. Therefore, it is necessary to take a general idea of the exceptional stability of BC in soil with some reserve [4,24].

On the other hand, it was found that the ratio of free light fraction and the occluded light fraction of soil organic matter correlates with BC content only in localities without deposition. In contrast, in localities with deposition, this relationship does not exist. This is confirmed by the conclusion of the previous paragraph and the finding that HBC is also bound in a light fraction of soil organic matter. As this ratio decreases in deposition sites, it is clear that the high sorption capacity of ABC leads to the formation of an occluded light fraction of oLFOM soil organic matter. It is a stabilizing mechanism [13]. Nevertheless, although fLFOM is more labile than oLFOM. However, oLFOM also mineralizes in the soil in the order of decades [12]. Therefore, ABC cannot be a pool of permanently stored carbon in the soil.

Correlation analysis revealed a close relationship between the BC content in the soil and the number of water-stable aggregates. In localities with atmospheric fallout deposition, this relationship is less significant. The formation of water-resistant macroaggregates is a much less sensitive indicator than the fLFOM/oLFOM ratio. Therefore, this result is not surprising; it only confirms the conclusion of the previous paragraph [50]. BC did not contribute to the higher forms of the occluded light fraction of organic matter within the aggregates and thus did not positively affect their amount [51]. This may be due to this phenomenon: BC, which has both sorption and ion exchange properties (HBC), is low in soil and is diluted in sites with ABC deposition (which has only sorption properties). Anthropogenic BC is not yet oxidized, having no negative charge network on its surface [29]. Therefore, there was no sorption of foreign, oxidized organic soil matter [33] and thus no increase in CEC [34]. The soils from both groups of localities had the same CEC. The dilution of "quality" HBC by "poor quality" ABC and other influences caused even higher CEC in soils from atmospherically cleaner localities, with lower BC content. Thus, the $C_{org}$ content in samples of localities with higher deposition of atmospheric fallout correlates more significantly (r = 0.840711) with CEC than in samples from localities without deposition (r = −0.310164).

### 3.3. Evaluation of Results from the Point of View Principal Component Analysis and Factor Analysis

On the graph of component weights PC1, PC2, PC3 (Figure 3), the first two axes are significant, which together exhaust about 92% of the variability. The PC1 axis in the PC1 × PC2 graph unambiguously characterizes WSM and BD, which go directly along this axis and are correlated with it at a level exceeding 0.81 and −0.80 (high correlation), as well as $C_{org}$ (r = 0.77) and particles 0.01 (r = −0.69). On the PC2 axis, there is a significant correlation between BC and DSS (r = 0.91 and 0.75). There is no significant correlation on the PC3 axis, but the direction is differentiated according to particles 0.01 (r = 0.63) and $C_{org}$ (r = 0.59).

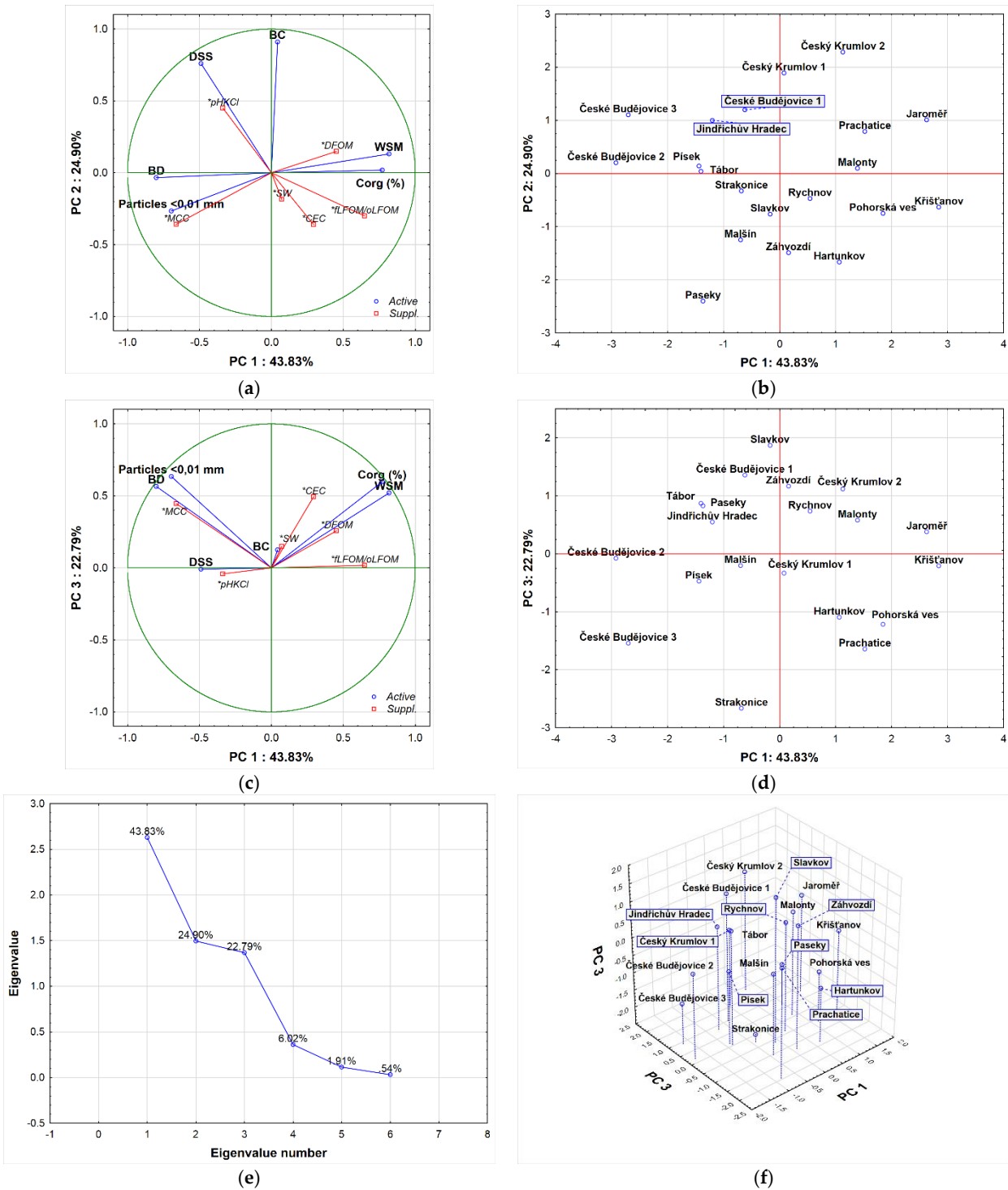

**Figure 3.** Principal component analysis (PCA) of soil quality parameters in the area of interest: (**a**) graph of component weights 1 a 2 (PC1 × PC2) for soil quality parameters. (**b**) The scatterplot of the component score PC1 × PC2 for individual localities. (**c**) Graph of component weights 1 a 3 (PC1 × PC3) for soil quality parameters. (**d**) The scatterplot of the component score PC1 × PC3 for individual localities. (**e**) Scree plot; (**f**) 3D graph PC1 × PC2 × PC3 for individual localities. Note: BC—black carbon (% $C_{org}$); fLFOM/oLFOM—the ratio of free light fraction and the occluded light fraction of soil organic matter; DFOM—the organic matter of the heavy soil fraction (% C of soil); WSM—water-stable macroaggregates (g aggregates (kg soil after sand subtraction)$^{-1}$); CEC—cation exchange capacity (mmol/kg); pH$_{KCl}$—soil reaction; particles <0.01 mm—clay particles; $C_{org}$—soil organic carbon (%); MCC—maximal capillary capacity (%); SW—specific weight (g/cm$^3$); DSS—degree of sorption saturation (%); BD—bulk density (g/cm$^3$).

In the scatterplot of the component score, sampling sites according to WSM, $C_{org}$, BD and particles are clearly located along the PC1 axis. The PCA divided the places in the area of interest into several different clusters. The cluster of localities České Budějovice 2 and 3 are characterized by a very high content of DSS, a higher BD and the lowest content of WSM and $C_{org}$ (also distinguished in the graph PC1 × PC3) and a higher content of BC. On the contrary, the localities Jaroměř and Křišťanov are characterized by the highest content of WSM, $C_{org}$ and the lowest BD and very similar content of particles. The Paseky and Hartunkov localities deviate from other localities with the lowest BC content (1.70–1.78%). The Český Krumlov 2 locality has the highest BC content (3.50%) with a high $C_{org}$ content (3.29%). Therefore, these two localities are at the most significant distance from each other within the PCA axis. Other localities have similar soil quality parameters and, thus, form the largest cluster around the center. The Strakonice locality with the PC1 × PC3 graph also differed significantly from other localities with a low content of BC (2.11%) and $C_{org}$ (2.34%).

Factor analysis (FA) (Figure 4) confirmed the results of PCA and differentiated similarly to the PCA method of the group of localities (see scattering diagrams of component scores). Factor weights explain the correlations between factors and features. They represent essential information on which the interpretation of factors is based. Factor 1 describes the properties in terms of $C_{org}$ and WSM. The cluster of localities České Budějovice 2, 3, and Strakonice are characterized by the lowest content of WSM and $C_{org}$, on the contrary, the localities Jaroměř and Křišťanov have the highest range of WSM, $C_{org}$. Factor 2 clearly describes the content of BC and DSS. The lowest BC content was determined in the Paseky and Hartunkov localities. Therefore, they are differentiated from other localities in the Factor 1 × Factor 2 graph, within the factor 2 axis. Factor 3 describes BD and particles. The Paseky locality has the highest BD and particles, while the Prachatice locality has the lowest BD and particles. Communality represents the proportion of character variability expressed by the factors in question. It is similar to the value of $R^2$, which we obtain when the original characters are explained by regression by selected factors [56]. From the contribution of Factor 1, Factor 2, and Factor 3 to communality, it is clear how communality acquires high values. Thus, the features of most values are very well considered by the proposed factor model (Table 1).

**Table 1.** Factor weights and contributions of a given factor to the communality for individual traits after rotation of varimax normalized soil quality parameters in the area of interest.

| Parameter | Factor Weight | | | Contributions of a Given Factor to the Communality | | | |
|---|---|---|---|---|---|---|---|
| | Factor 1 | Factor 2 | Factor 3 | Factor 1 | Factor 2 | Factor 3 | Communalities |
| BC | 0.1885 | 0.8944 | −0.1017 | 0.0355 | 0.8355 | 0.8458 | 0.4384 |
| BD | −0.1917 | 0.1561 | 0.9496 | 0.0368 | 0.0611 | 0.9628 | 0.9087 |
| DSS | −0.2992 | 0.8325 | 0.1843 | 0.0895 | 0.7827 | 0.8166 | 0.5265 |
| $C_{org}$ | 0.9692 | −0.0750 | −0.0854 | 0.9393 | 0.9449 | 0.9522 | 0.8948 |
| WSM | 0.9578 | 0.0212 | −0.1946 | 0.9173 | 0.9178 | 0.9556 | 0.9084 |
| Particles < 0.01 mm | −0.0876 | −0.0849 | 0.9715 | 0.0077 | 0.0149 | 0.9586 | 0.8845 |

Note: BC—black carbon (% $C_{org}$); BD—bulk density (g/cm$^3$); DSS—degree of sorption saturation (%); $C_{org}$—soil organic carbon (%); WSM—water-stable macroaggregates (g aggregates·(kg soil after sand subtraction)$^{-1}$); particles < 0.01 mm—clay particles (%).

Multidimensional statistical methods (multicriteria evaluation using PCA and FA methods) significantly enabled, based on extensive data analysis, to differentiate the area of interest in the evaluated parameters (WSM, $C_{org}$, BC, etc.) into different clusters (individual localities within differentiated clusters have their own qualitative parameters soils very similar).

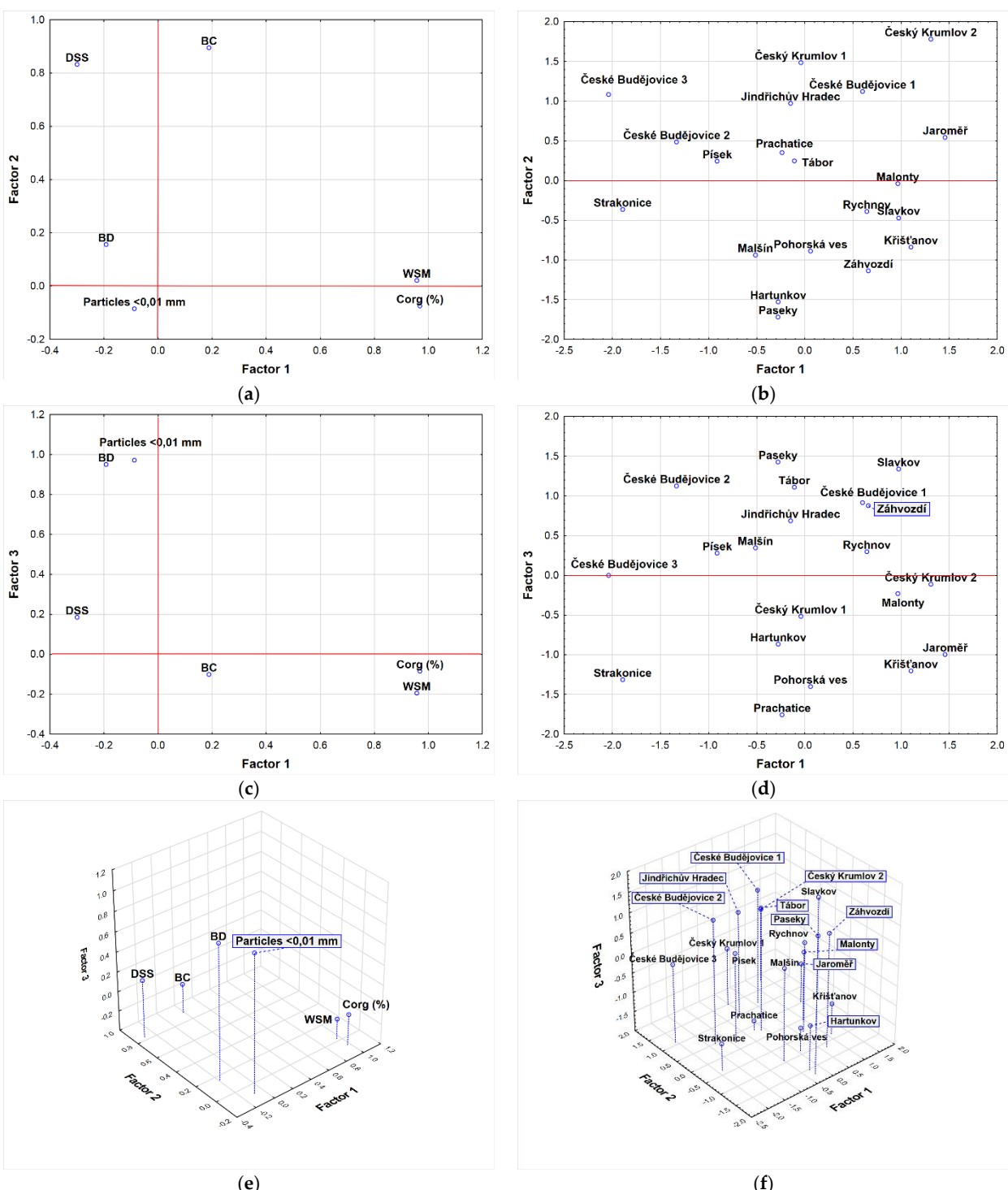

**Figure 4.** Factor analysis (FA) of soil quality parameters in the area of interest: (**a**) graph of factor weights (Factor 1 × Factor 2) for soil quality parameters. (**b**) Scattering diagram of component scores Factor 1 × Factor 2 for individual localities. (**c**) Graph of factor weights (Factor 1 × Factor 3) for soil quality parameters; (**d**) scattering diagrams of component score Factor 1 × Factor 3 for individual localities; (**e**) 3D graph Factor 1 × Factor 2 × Factor 3 for soil quality parameters; (**f**) 3D graph Factor 1 × Factor 2 × Factor 3 for individual localities. Note: BC—black carbon (% $C_{org}$); fLFOM/oLFOM—the ratio of free light fraction and the occluded light fraction of soil organic matter; WSM—water-stable macroaggregates (g aggregates·(kg soil after sand subtraction)$^{-1}$); particles < 0.01 mm—clay particles; $C_{org}$—soil organic carbon (%); DSS—degree of sorption saturation (%); BD—bulk density (g/cm$^3$).

## 4. Conclusions

Organic matter influences the soil ecosystem and, indirectly, the environment. One of the important fractions of soil organic matter is black carbon. Therefore, this study investigated black carbon in the clay Cambisols with its different atmospheric fallout. On average, lower black carbon content was found in sparsely populated areas of southern Bohemia (2.16% $C_{org}$). In comparison, it was higher in areas of larger cities with a significant anthropogenic load (2.76% $C_{org}$). Black carbon in the soils of larger cities probably has an anthropogenic origin. Thus, two types of black carbon were found. The first one is historical from biomass fire when the second one is anthropogenic and relatively new. It is created from combustion in furnaces and transport fumes, and does not significantly affect organic matter. However, both groups of black carbon have entirely different properties and effects on the environment. Historical black carbon from atmospherically cleaner localities is firmly bound with mineral soil colloids. It is evident from the relationship of its content in the soil to the heavy fraction of soil organic matter; moreover, from the ratio of free light fraction and the occluded light fraction of soil organic matter. On the contrary, in soils from localities loaded by atmospheric fallout, the relationship of black carbon content to these parameters is negligible. Thus, it can therefore be stated that anthropogenic black carbon is much less stable in soils than historical black carbon because it has not yet been associated with mineral colloids. This is because its chemical properties are entirely different from historical black carbon or has not been oxidized in soils yet. That is why anthropogenic black carbon has only the sorption capacity, not the cation exchange capacity, in contrast with historical black carbon. The high sorption capacity of anthropogenic black carbon leads to forming an occluded light fraction of soil organic matter, which is a stabilizing mechanism, although very slow. The free light fraction of soil organic matter is less stable than the occluded fraction. However, even the occluded fraction mineralizes in the soil for decades, so anthropogenic black carbon cannot be a pool of permanently stored carbon in the soil.

**Supplementary Materials:** The following are available online at https://www.mdpi.com/article/10.3390/agronomy11112261/s1, Table S1: Basic descriptive statistics; Figure S1: The map of sampling localities (South Bohemia, Czech Republic).

**Author Contributions:** Conceptualization, M.K. and P.M.; methodology, L.K.; formal analysis, J.B.; investigation, T.N.H.; data curation, L.M.; writing—original draft preparation, L.K.; writing—review and editing, M.K.; supervision, M.D.; project administration, R.V.; funding acquisition, P.K. All authors have read and agreed to the published version of the manuscript.

**Funding:** This work was supported by the University of South Bohemia in České Budějovice (no. GAJU 059/2019/Z) and the research plan of the Ministry of Agriculture of the Czech Republic-RO0418.

**Institutional Review Board Statement:** Not applicable.

**Informed Consent Statement:** Not applicable.

**Conflicts of Interest:** The authors declare no conflict of interest.

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
