# Peer review of "Black Carbon and Its Effect on Carbon Sequestration in Soil"

_agronomy, doi:10.3390/agronomy11112261_

Round 1

Reviewer 1 Report

  1. The question posed by the authors is new and it is well defined.
  2. The method is appropriate and well described There are sufficient details and it is possible to replicate it.
  3. The data sound and well controlled, 
  4. The figures appear to be genuine.
  5. The manuscript adд here to the relevant standards for reporting and data deposition
  6. I propose the authors to add some more explanation of the results in the conclusions
  7. The title and abstract are correct, accurately convey what has been found
  8. The writing is acceptable
  9. Somebody with mater English to look over the manuscript

Author Response

Dear reviewer,
Thank you very much for your time and for writing a review. Based on your recommendation, we have revised the Conclusion to reflect the new findings described in the article.

Sincerely,

Marek Kopecký et al.

Reviewer 2 Report

Kopecký et al., have studied the properties of black carbon in the clay cambisol in order to determine if carbon sequestration in the form of black carbon is more or less significant in the given soil and climate conditions. I consider that the article is interesting because they find that in the studied soils there are two types of black carbon (historical from biomass fire and anthropogenic). They fount that the properties of these two types of black carbon have completely different effect on the environment because anthropogenic black does not improve soil quality as much as historical black carbon from biomass.

I consider that the methodology followed is adequate. However, I would like to suggest the authors several modifications:

  • In the section Materials and methods, it is crucial to include a map in the section soil samples processing. The authors give a description of the selected area but without a map it is difficult to imagine the área
  • In the same section, it should be clarified if the samples have been taken in triplicate and the analyzes have been done in triplicate, since this is an important piece of information for the subsequent statistical study.
  • Improve figures 1 and 2 because the error bar is not clearly visible. In addition, it should be indicated in these graphs if there are significant differences (n = 3; P < 0.05). So, analysis of variance (ANOVA) should be include to determine differences between samples. The results of this analysis can be represented in figure 1 and 2.

Author Response

Dear reviewer,
Thank you very much for your time and for writing a review.
Following your recommendations, we have improved the submitted article. A locality map has been added to the supplements. The GPS coordinates were added to Table S1. The figures have also been improved according to your recommendations. 

With respect,
Marek Kopecký et al.